# On the Innovation Design for Two-Motor Transmissions with Eight-Link Mechanisms in the Electric Vehicles

**Ngoc-Tan Hoang**  **and Hong-Sen Yan** *

Department of Mechanical Engineering, National Cheng Kung University, 1 University Road, Tainan 70101, Taiwan; hoangngoctan@iuh.edu.vn
* Correspondence: hsyan@mail.ncku.edu.tw; Tel.: +886-0939-515-000

**Abstract:** A decade ago, electric vehicles (EV) made a boom in the automobile market, as they started to become a growing market section in the transportation space. The reasons behind the boom were to decrease environmental pollution by reducing the use of fossil fuels, lowering transportation operating costs, and increased general consumer interest in the new technology. This work generates a streamlined process for the design and simulation of motor transmissions with eight-link mechanisms. This procedure presents a wide range of motor transmissions such as 34 new clutchless systems and 34 new clutched systems. Two novel feasible motor transmissions of the design process are taken as a sample to dissect the working principle conjoined both power flow paths and operation modes. In addition, these designs are conducted for modeling and computer simulation procedures that obtain the results of the energy management strategy and operation mode variation.

**Keywords:** design process; motor transmission; electric vehicles; operation modes; eight-link mechanism; transmission simulation

## 1. Introduction

In the mid-19th century, the first electric vehicle powered with dischargeable original cells was created by Robert Anderson [1]. Today, many researchers and companies present their designs [2–11]. Based on the development history of electric vehicles, four main types of EV architectures are popular today [12], namely, central dive, drive derivatives, wheel-hub drive, and wheel-hub drive derivatives. Figure 1 shows the central dive architecture that has been selected to as the design and simulation model used in this paper.

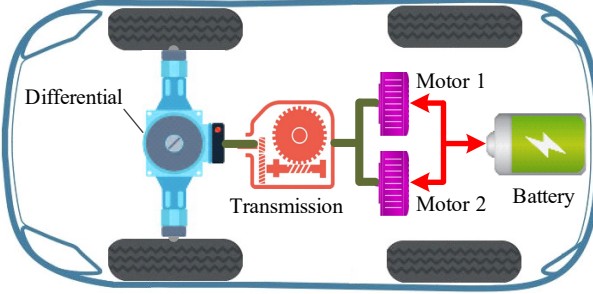

**Figure 1.** The central dive motor transmission of an electric vehicle (EV).

In 1996, Schimidt [13] improved a transmission having two planetary gear trains (PGT). Both the Sun gears and Ring gears of each PGT are connected together. The output shaft is operatively conjoined

to the differential and the carrier in the first PGT. The system uses one clutch and one brake for the transferred operation modes.

Tamai et al. [14] took a patent of the multispeed drive transmission system in 2013. Compound-input is provided for an electrically variable transmission for an electric vehicle. The compound-input electrically variable transmission has improved input gear ratios that allow the motors to be operated in its desired efficiency and/or performance range during both city and highway vehicle operation. Further, the multispeed input electrically variable transmission provides an input brake without the need for a dedicated input brake clutch or braking mechanism and incorporates a reverse gear for reverse operation.

In 2015, Holmes et al. [15] presents an electrically variable transmission with a compound PGT that may be only two PGTs. Two members of the first PGT are conjoined for common rotation with two members of the second PGT. A first motor/generator is conjoined for common rotation with a member of the first PGT and a member of the second PGT. An input member is conjoined to another member of the first PGT. The first set of intermeshing gears includes a first gear conjoined for common rotation with one of the gears of the second PGT and a second gear driven by a second motor/generator. A second set of intermeshing gears includes another gear driving the output member. The system uses two clutches for the transferred operation modes. The GM also used this patent and improved their vehicles through its application.

With the innovation of Yan's methodology for the creative design [16], Hoang and Yan synthesized and analyzed the feasible configurations of novel transmissions with 12 joints and 8 members [17,18]. Depending on the method, this paper proposes an approach for the synthesis, computer simulation, and modeling of central drive motor transmissions with eight-link mechanisms (shown in Figure 2).

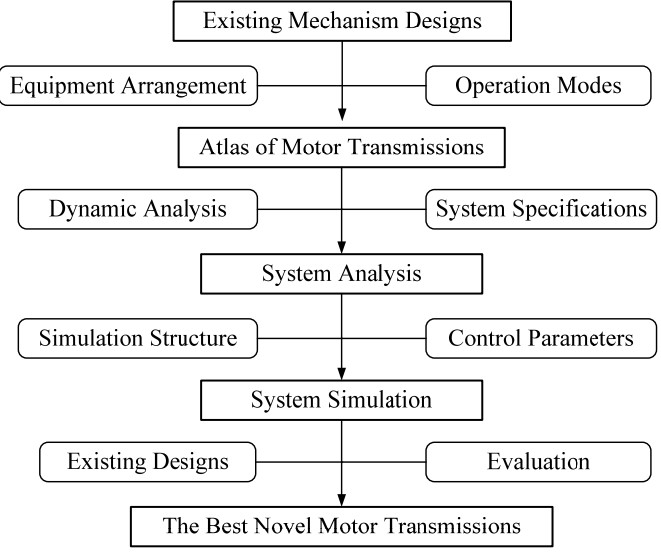

**Figure 2.** The design procedure.

## 2. Existing Mechanism Designs

Figure 3 shows the summarized process of the configuration synthesis for eight-link mechanisms in the novel series-parallel hybrid transmissions [17]. The corresponding feasible specialized chains are detected with three steps:

Step 1: Allocate links. Based on the design conditions such as requirements and constraints, links are allocated to each generalized kinematic chain (see Figure 3(a1)).

Step 2: Allocate joints. The gear joints and the revolute joints are allocated to the results in Step 1 (see Figure 3(a2)).

Step 3: Transfer diagrams. The specialized chains are transformed into the block diagrams in the results in Step 2 (see Figure 3(a3)).

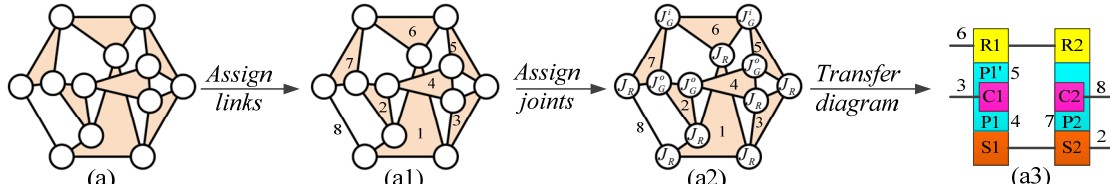

**Figure 3.** Process of specialization. (**a**) generalized kinematic chain; (**a1**) specialized chain with identified all links; (**a2**) specialized chain with identified all joints; (**a3**) block diagram.

Based on these results, 18 feasible mechanisms are selected for the design and simulation of motor transmissions with eight-link mechanisms in EV, Figure 4.

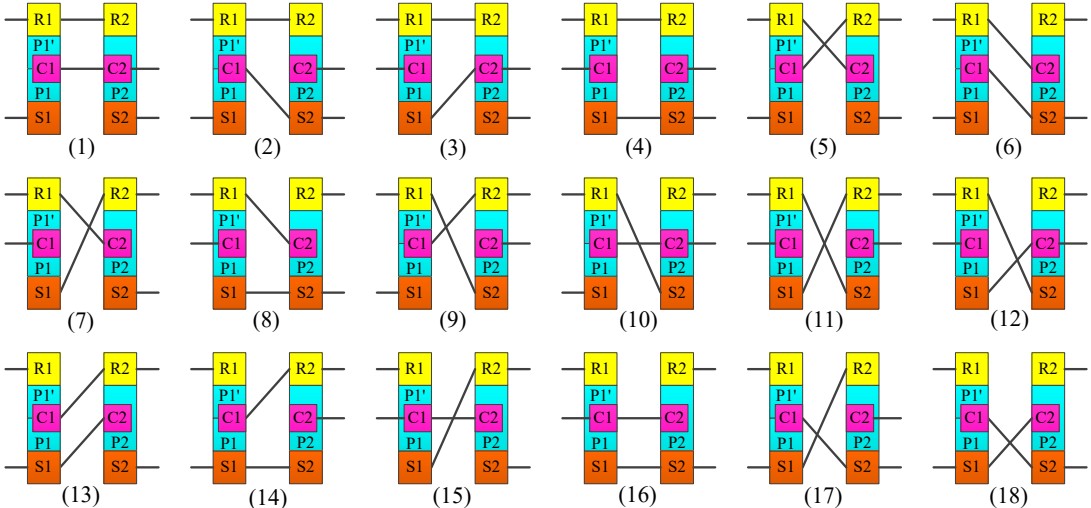

**Figure 4.** Atlas of eight-link mechanisms of motor transmissions.

## 3. The Atlas of Motor Transmissions

By selecting the mechanisms above, the clutch arrangement, the power arrangement, and the operation modes will be considered here.

### 3.1. Power Arrangement

With the obtained mechanisms, the configuration described in Figure 4(4) is used in an attempt to verify this process. With the design constraints of the inputs/output, the output and motors are allocated to each mechanism with three main steps:

Step 1: Allocate the Output (O). The O shaft will be conjoined to one member of compound PGT as the carrier or the ring gear in order to upgrade high torque of the output. In the compound PGT, if the third link is held, the O shaft will be reduced the speed than the motor speed. Two results have been created, as shown in Figure 5(4.1,4.2).

Step 2: Allocate the Motor 1 (M1). The M1 shaft will be conjoined to one member of the compound PGT as the sun gear or the ring gear in order to obtain higher torque and reduce the speed for the O shaft, the M1 should not be conjoined to the carrier to avoid excessive output speed. The M2 speed should be reduced the speed than the speed of M1. The assignment of the M1 creates two results (see Figure 5(4.3,4.4)).

Step 3: Assign the Motor 2 (M2). The M2 shaft will be conjoined to one member of compound PGT as the carrier or the ring gear in order to have high speed of the vehicle. The assignment of the M2 creates two results (see Figure 5(4.5,4.6)).

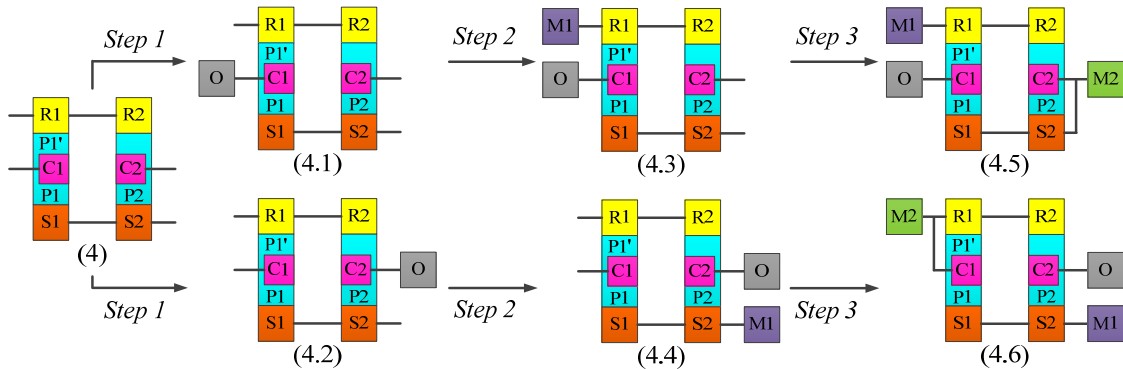

**Figure 5.** The procedure of power arrangement.

When applying this process for 18 mechanisms, the atlas of clutchless motor transmissions including 34 systems is generated and shown in Figure 6.

**Figure 6.** Atlas of the new motor transmissions without clutch.

## 3.2. Clutch Arrangement

In this section, brakes and clutches are added to the configuration to adjust its operation modes to follow the demanded operation modes. The required operation modes and clutch arrangement are described in two basic steps, as shown below.

### 3.2.1. Required Operation Modes

Since the purpose of this work is to develop new motor transmissions, the operation modes of new configurations are selected depending on the operating conditions of the vehicle, stated by Mi et al. [19]:

- M1 mode: the M1 alone drives the vehicle to provide a high torque for the vehicle in the start running state or the climbing hill state.
- M2 mode: the transmission transfers to the M2 mode to improve efficiency when the vehicle demands a moderately high power (highway cruise control).
- Combined power mode: the transmission transfers to the power mode to provide more power when the vehicle has a heavy load status or high-speed acceleration status. The power mode is the combination of M1 power and M2 power.
- Regenerative braking mode: In order to store the electric in the batter, the motors will work as a generator. In the vehicle's braking status, the kinetic energy turns into electric.

### 3.2.2. Clutch Arrangement

With the number of operation modes and motor statuses in each mode, the minimal number of brakes and clutches are added to each configuration to achieve the demanded operation modes. The systems described in Figure 6(6) and Figure 7 are used as samples to verify the clutch arrangement procedure (see Figure 7).

Step 1: M1 mode. Since the M1 mode is demanded, the system is conducted with a 1-DoF configuration while the M1 drives the vehicle. Brakes and clutches are added to generate each result, as shown in Figure 7(6.1,7.1).

Step 2: M2 mode. Since the M2 mode is demanded, the system is still conducted with a 1-DoF configuration while the M2 drives the vehicle. Brakes and clutches are added to generate each result, as shown in Figure 7(6.2,7.2).

Step 3: Combine mode. Both motors will be conjoined to the O shaft; therefore, the systems transfer to a 2-DoF, as shown in Figure 7(6.3,7.3).

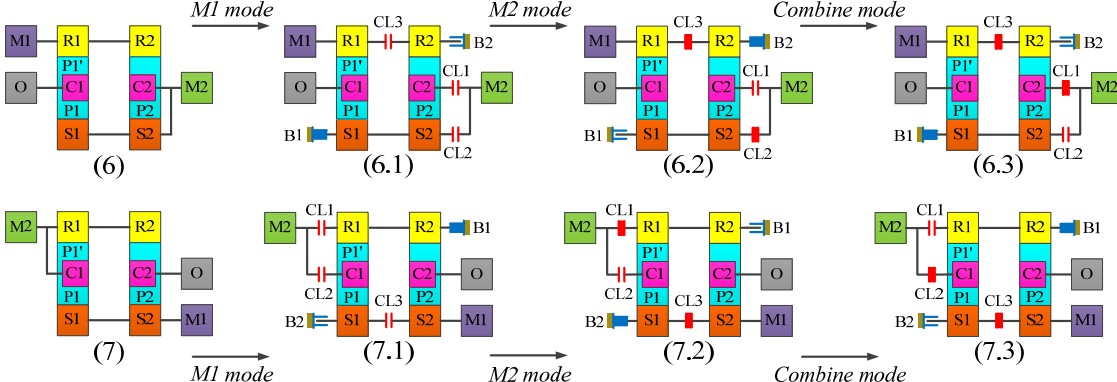

**Figure 7.** The procedure of clutch arrangement.

As the result, there are 34 clutched systems generating the new corresponding motor transmissions by applying this process to all configurations (see Figure 8).

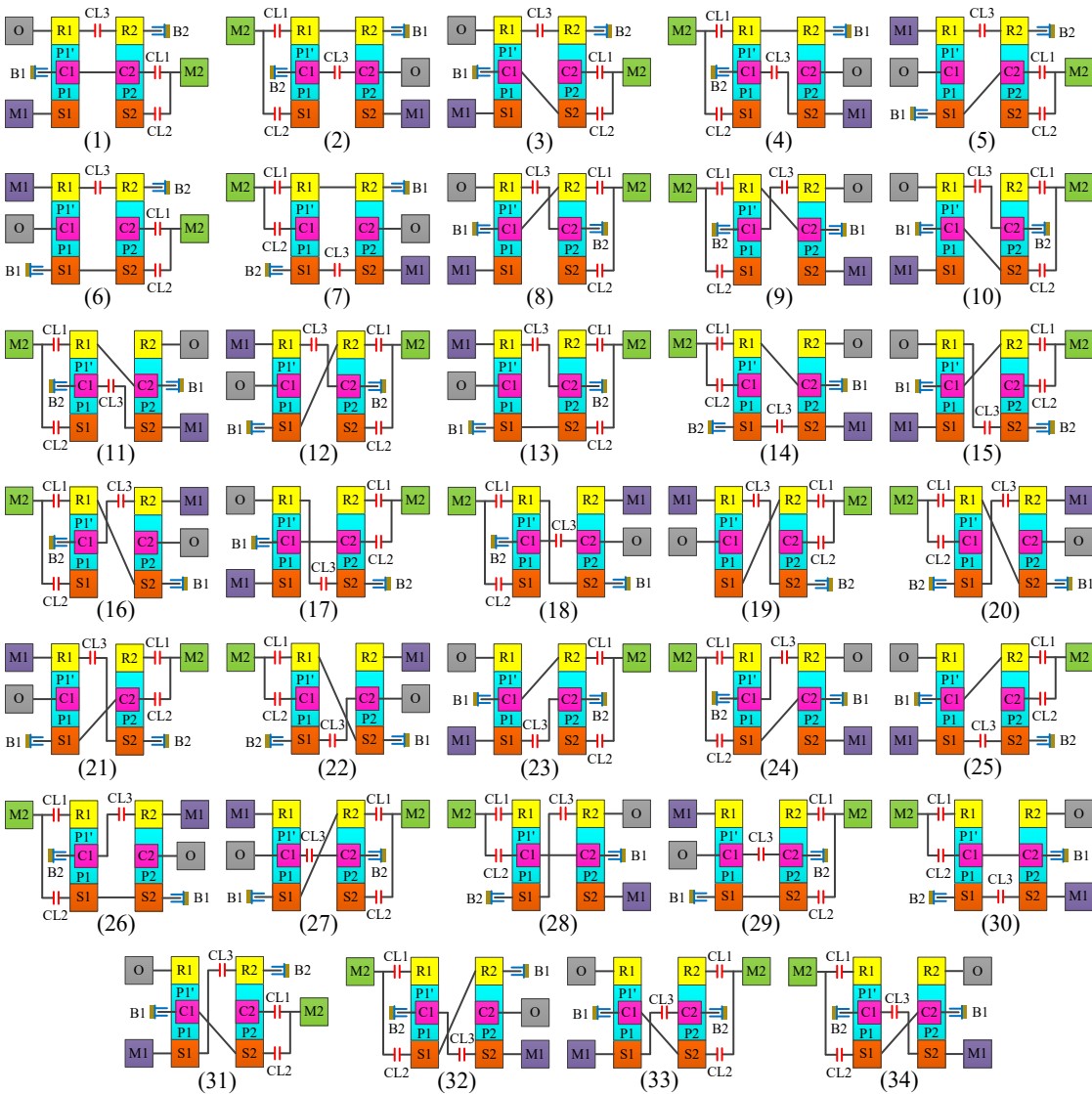

**Figure 8.** The atlas of the new motor transmissions.

## 4. System Analysis

The modeling to determine the specifications, operation modes, and functions for the simulation procedure is shown in this section, including the system specifications and the dynamic analysis.

### 4.1. The Dynamic Analysis

Another way to verify the reasonability of the newly-synthesized motor transmissions is by using the design configurations shown in Figures 7 and 8(6); they configurations are selected as the examples to describe either power flow analysis or operation modes of the new systems (see Figure 9). Figure 10 presents the operation modes distributed into sixteen reasonable clutching conditions following in Table 1 (an "x" represents the engagement of clutches or brakes.

In each operation mode, the correlation between the torque and the power provides a reference value of power. In specificity, the reasonable operation modes are determined based on the demanded vehicle power. Moreover, operation modes will be conducted to donate the flexible control to EVs during operation and improve energy economy by adding more brakes and clutches to the transmissions.

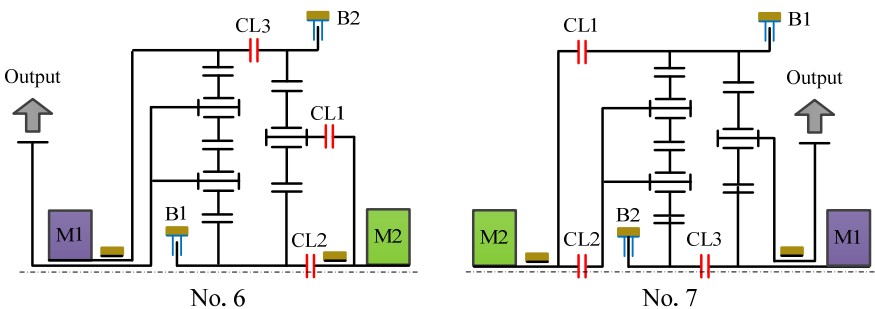

**Figure 9.** The new motor transmissions.

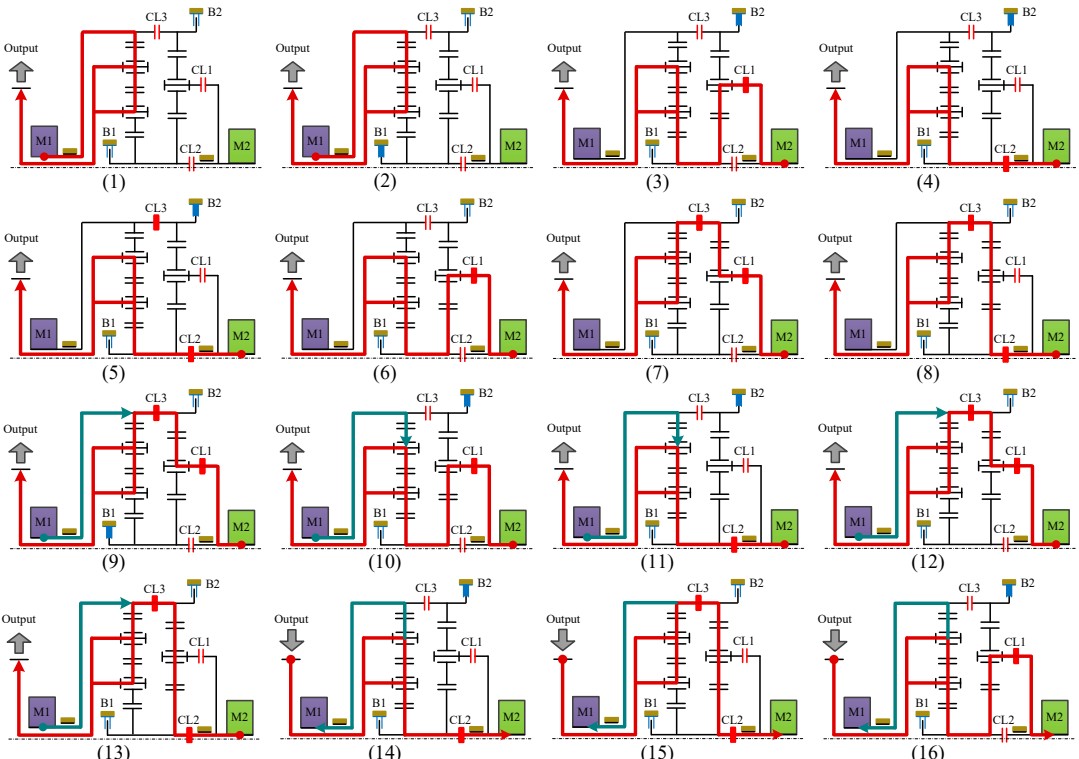

**Figure 10.** Operation modes for the No. 6 motor transmission.

**Table 1.** Operation modes and clutching conditions.

| No. | Operation Modes | No. 6 | | | | | | No. 7 | | | | | |
|---|---|---|---|---|---|---|---|---|---|---|---|---|---|
| | | Clutch and Brake Engagement | | | | | Reduction Ratio | Clutch and Brake Engagement | | | | | Reduction Ratio |
| | | C1 | C2 | C3 | B1 | B2 | | C1 | C2 | C3 | B1 | B2 | |
| 1 | Motor 1-1 | | | | | | 1.3 | | | | | | 1.3 |
| 2 | Motor 1-2 | | | | x | | 3 | | | | x | | 3 |
| 3 | Motor 2-1 | x | | | | x | 1 | x | | | | x | 1.3 |
| 4 | Motor 2-2 | | x | | | x | 3 | | x | | | x | 1 |
| 5 | Motor 2-3 | | x | x | | x | 3 | | x | x | x | | 0.8 |
| 6 | Motor 2-4 | x | | | | | 1.8 | x | | x | | x | 1.8 |
| 7 | Motor 2-5 | x | | x | | | 1 | x | | | | | 1.3 |
| 8 | Motor 2-6 | | x | x | | | 0.8 | | x | | | | 3 |
| 9 | Combined power 1 | x | | x | x | | 0.8 | | x | x | x | | 0.8 |
| 10 | Combined power 2 | x | | | | x | 1 | x | | | | x | 1.8 |
| 11 | Combined power 2 | | x | | | x | 3 | | x | | | x | 1 |
| 12 | Combined power 4 | x | | x | | | 0.8 | x | | x | | | 1.3 |
| 13 | Combined power 5 | | x | x | | | 1.3 | | x | x | | | 3 |
| 14 | Regenerative braking 1 | | x | | | x | 3 | | x | x | | | 3 |
| 15 | Regenerative braking 2 | | x | x | | | 1.3 | x | | x | | | 1.8 |
| 16 | Regenerative braking 3 | x | | | | x | 1 | x | | | | x | 1 |

*4.2. System Specifications*

This part shows the demanded system specifications as power sources specifications, vehicle specifications, desired performances, and environment conditions (see Table 2).

**Table 2.** The system specifications.

| Vehicle | | Motor 1 | |
|---|---|---|---|
| Weight *(m)* | 1.670 kg | Model | Westinghouse 75 |
| Frontal area $(A_f)$ | 2.20 m² | Mass | 91 kg |
| Wheel radius $(R_w)$ | 0.33 m | Type | AC Induction |
| Rolling resistance coefficient $(C_r)$ | 0.02 | Maximum Speed | 10,000 rpm |
| Transmission efficiency $(\eta_t)$ | 0.95 | Maximum power | 55 kW |
| **Driving environment** | | Maximum torque | 271 Nm |
| Incline angle $(\theta)$ | 0 degree | Efficient region | 3000–10,000 rpm |
| Gravity $(g)$ | 9.81 m/s² | Peak efficiency | 92% |
| Air mass density $(\rho)$ | 1.225 kg/m³ | **Motor 2** | |
| Aerodynamic drag coefficient $(C_d)$ | 0.35 N*s²/kg*m | Model | Prius Japan |
| **Performance** | | Mass | 76 kg |
| Maximum speed | 200 km/h | Type | AC Induction |
| Acceleration time | 9 s | Maximum Speed | 6000 rpm |
| Grade ability | 32% | Maximum power | 75 kW |
| **Battery** | | Maximum torque | 322 Nm |
| Model | MY 2016 Slithium-ion battery | Efficient region | 2000–6000 rpm |
| Max capacity $(C_N)$ | 85 kWh | Peak efficiency | 90% |

## 5. System Simulation

With the new motor transmissions, the computer simulation will be built in this part by using MATLAB/SIMULINK (R2017b, The MathWorks, Inc., East Natick, MA, USA).

*5.1. Simulation Structure*

The simulation model is developed including the vehicle systems, the driving force calculation, the drive cycle, the control logic, the transmission, and the power sources system, as shown in Figure 11. The simulation results of the new design are regarded to as the New European Drive Cycle (NEDC).

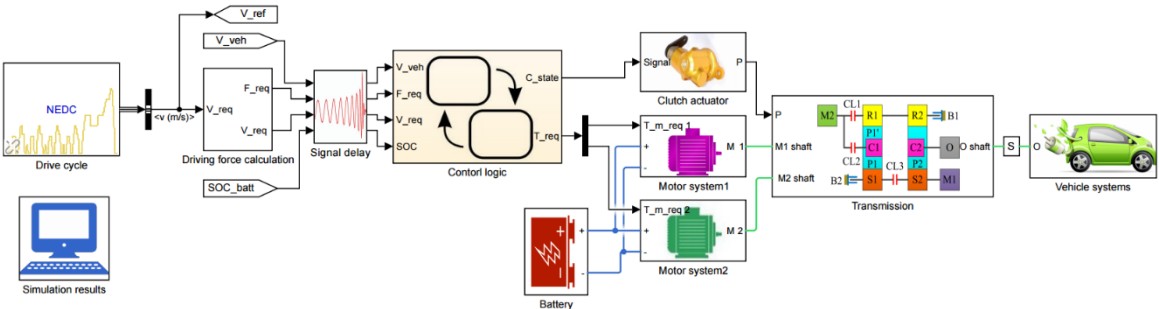

**Figure 11.** The simulation structure.

*5.2. Control Parameters*

Through the gear definition procedure, a reasonable gear is chosen according to the vehicle speed; furthermore, both the minimum energy consumption of every 1-DoF operation mode and the reduction ratio are also calculated. After the vehicle driving conditions and the energy consumptions

are predicted, the mode definition procedure is organized to choose the most reasonable operation modes. Ultimately, the power source need generate the optimized torque and the clutching signal are distributed to the motor transmission.

With the high-efficiency region, the gear is still chosen to keep the motor operating in the gear definition procedure; hence, the reasonable reduction ratio is obtained at that moment (see Figure 12). The state of gear will be zero when the demanded driving force is zero and the vehicle stops; otherwise, the state of gear will be a positive integer. When the gear threshold speeds are equal to the vehicle speed, the state of gear will change to maintain the high motor efficiency.

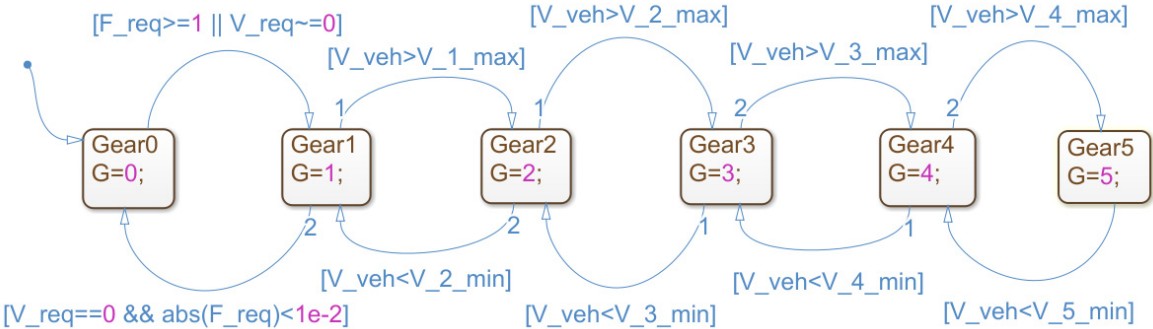

**Figure 12.** The gear definition control.

Three basic states—the driving state, the regenerative braking state, and the parking state—are decided in the mode definition procedure, as shown in Figure 13. The reasonable operation mode will be available in each state including the stopping vehicle uses the parking state, the moving vehicle uses to the driving state, and the regenerative braking mode is chosen in case of the negative torque.

The vehicle begins with the M1 mode in the driving states. With the feasible shifting analysis, a reasonable mode in several modes is chosen in the same gear. Therefore, the vehicle leaves the starting state and transfers to the normal driving states, the operation mode will be taken according to the evaluated energy consumptions.

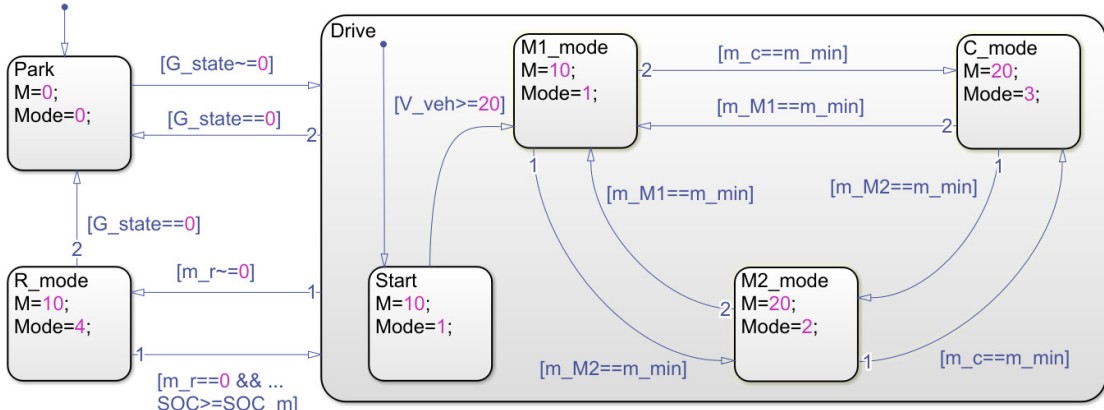

**Figure 13.** The state flow chart of mode definition control. G_state: the state of gear; m_M1: mass flow rate of Motor 1; m_M2: mass flow rate of Motor 2; m_min: minimum mass flow rate; m_c: mass flow rate of compound mode; m_r: mass flow rate of the regenerative braking mode.

### 5.3. The Simulation Results

The computer simulation results of two designs following to the driving cycle shown in Figures 14 and 15. The results assess the transition of energy consumption, the operation modes, and the gear states in the new European driving condition.

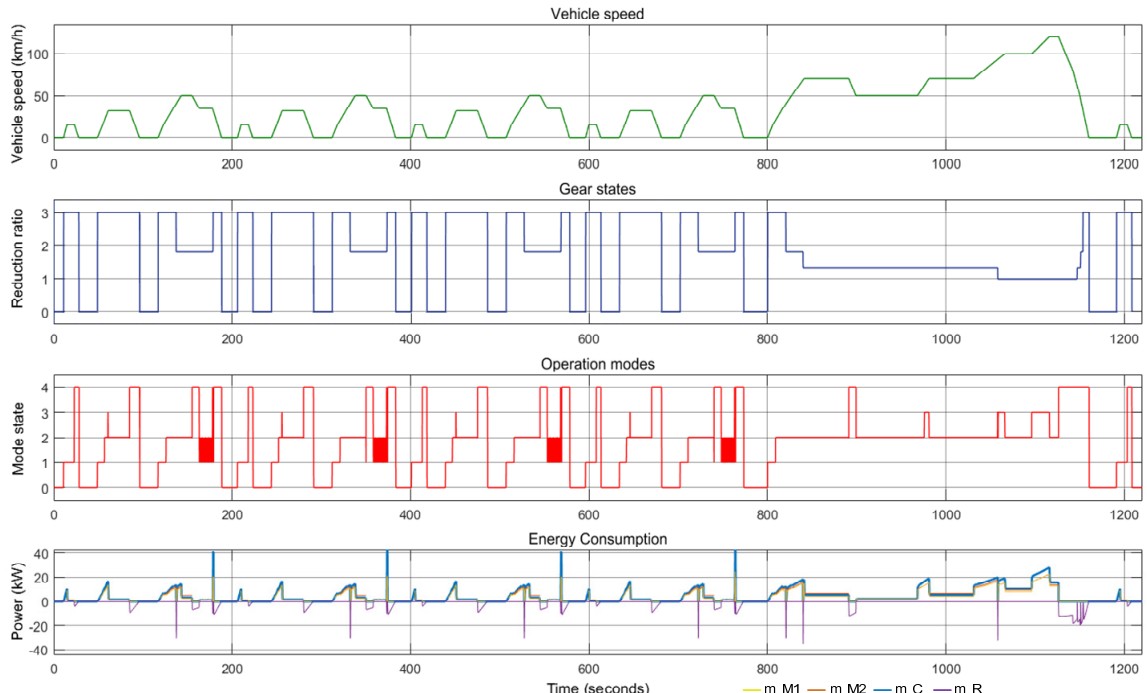

**Figure 14.** The simulation results of No. 6.

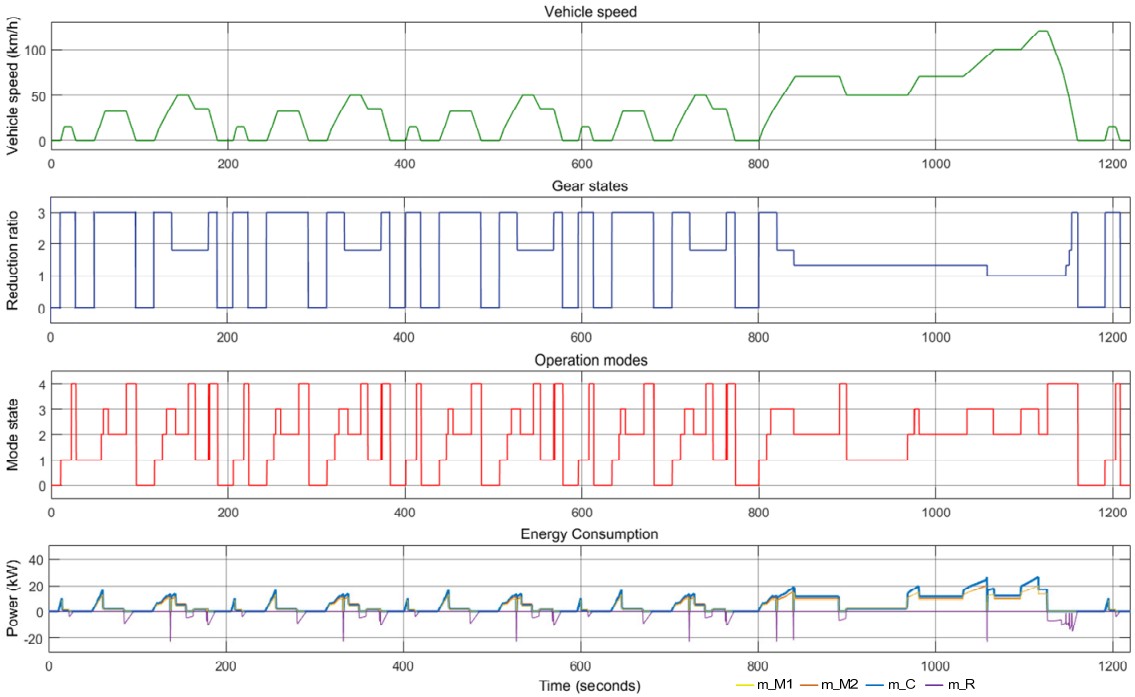

**Figure 15.** The simulation results of No. 7.

A comparison between No. 6 and No. 7 for the operation modes transition is shown on Table 3 and Figure 16. The percentage and time of operation modes hold on the driving cycle present that the results are suitable.

From the comparisons of the No. 6 and No. 7 results, the converted mode states of No. 7 are suitable for the driving cycle. The results are equally changed among the motor 1 mode, motor 2 mode, parking mode, regenerative braking mode, and combined power mode. Therefore, the No. 7 motor transmission operates more smoothly than the No. 6 motor transmission.

**Table 3.** The comparison No. 6 and No. 7 for the operation mode selection.

| Operation Modes | Percentage (%) | | Time (s) | |
|---|---|---|---|---|
| | No. 6 | No. 7 | No. 6 | No. 7 |
| 1. Parking | 22.8 | 22.8 | 278 | 278 |
| 2. Motor 1 | 13.7 | 23.6 | 167 | 288 |
| 3. Motor 2 | 44.8 | 26 | 547 | 317 |
| 4. Combined power | 3.1 | 12 | 38 | 147 |
| 5. Regenerative braking | 15.6 | 15.6 | 190 | 190 |
| Total | 100 | | 1220 | |

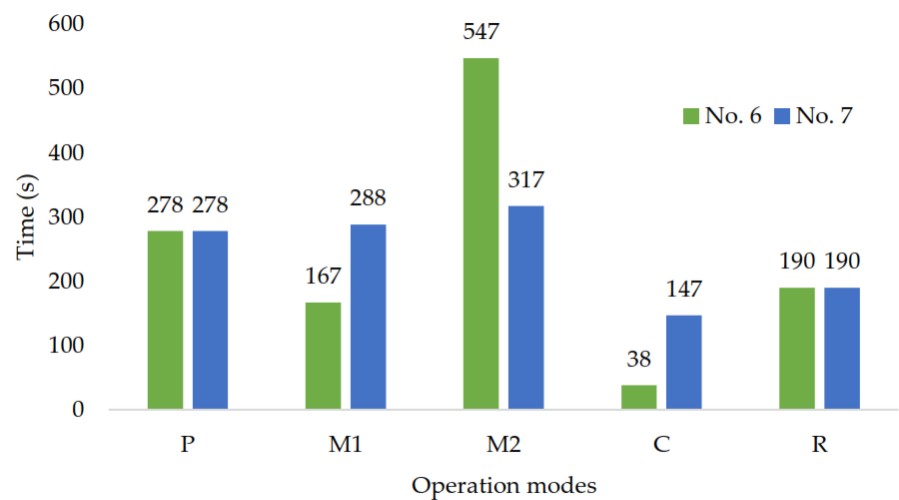

**Figure 16.** The comparison of operation mode selection between No. 6 and No. 7.

## 6. Conclusions

Based on the available results of the eight-link mechanisms, they are selected to allocate clutches and brakes by using the techniques of clutch arrangement and power arrangement. As a result, 34 clutchless and 34 clutched motor transmissions are created. Besides analyzing two new motor transmission systems, two computer simulation models are also constructed to verify the transition in the operation modes for the NEDC driving cycle. Besides, the variety range of motor transmissions and mechanisms is determined by either the desired operation modes or design constraints, the different systems can be also generated from a broad range of the taken operation modes offering a flexible description for engineers. Based on this design procedure, a total of the best motor transmissions is identified by synthesizing, analyzing, and simulating many motor transmissions.

**Author Contributions:** Data curation, N.-T.H.; Funding acquisition, H.-S.Y.; Methodology, H.-S.Y.; Writing—original draft, N.-T.H.; Writing—review & editing, H.-S.Y.

**Funding:** This research was funded by The Ministry of Science and Technology of Taiwan (MOST) grant number MOST 104-2221-E-006-059-MY3.

**Acknowledgments:** The Ministry of Science and Technology of Taiwan (MOST, R.O.C.) supported this work.

**Conflicts of Interest:** Hoang and Yan declare no conflict of interest.

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
