# Peer review of "On the Innovation Design for Two-Motor Transmissions with Eight-Link Mechanisms in the Electric Vehicles"

_applsci, doi:10.3390/app9010140_

Reviewer 1 Report

Authors present a design procedure regarding the motor transmission of an Electric Vehicle. The novelty of the proposed design is not clear, so authors have to elaborate on this. In addition, simulation results are not sufficient to validate the proposed design; additional results should be presented for competitive transmission system design, supporting an insightful comparison.

Author Response

We have attack the replied file below

We are very pleasure to receive these comments from you, if you have any comments regarding the revision of our manuscript, please fulfill to contact us. Thank you very much for your comments.

Reviewer 2 Report

This manuscript introduced a good design research work, but which can be improved in terms of the following fields:

Evaluation analysis should be an important part of the synthesis work too as well as the work in section 3. Here,based on simulation results, only the configurations are evaluated according to the average usage of their operating modes described in Row 211-214, while not drivability,economy,efficiency,regenerative efficiency and so on. That is not the key evaluation index. 

In Figure 14, there are three confusing questions. 1) the data from the simulated operation modes curve is only 0\1\2\3\4, which obviously does not come from Table 1(it sould be but not), or them comes from the txt in row 107-115. Please give their meaning for the numbers.2) it can be found in many field that the reduction ratio does not change while mode changes, the reduction ratio changes while mode doest not changes. For example, between 150s-180s and 850s-1000s and many others. Please check your results. 3) High-frequency mode changement between mode 1 and 2 is found. that should be avoided and improved by optimizing the gear structural or motor parameters.    

In Table 1, six gears for M2 mode and 5 gears for C Mode are listed. However, not all gears are necessary for EVs. Why not do some selection work?

In Figure 13, there are many strings, such as G_state, m_M1, m_M2,m_min, m_c, m_r, SOC_m, which are not explained in the context.

The synthesis work is suitalbe only for two-motor transmissions in EVs. The manuscript title is suggested to show this detailed information. 

Author Response

We have attach the replied file below

We are very pleasure to receive these comments from you, if you have any comments regarding the revision of our manuscript, please fulfill to contact us. Thank you very much for your comments. 

Round  2

Reviewer 1 Report

Authors have responded to the reviewers' comments, but they cannot provide experimental validation.